

# Capturing complexity: field-testing the use of 'structure from motion' derived virtual models to replicate standard measures of reef physical structure

Daniel T.I. Bayley[1,2,3], Andrew O.M. Mogg[4,5], Heather Koldewey[3,6] and Andy Purvis[1,7]

[1] Department of Life Sciences, Natural History Museum of London, London, UK
[2] Centre for Biodiversity and Environment Research, University College London, University of London, London, UK
[3] Conservation Programmes, Zoological Society of London, London, UK
[4] Tritonia Scientific, Oban, UK
[5] NERC National Facility for Scientific Diving, Scottish Association for Marine Science, Oban, UK
[6] Centre for Ecology and Conservation (CEC), University of Exeter, Penryn Campus, Cornwall, UK
[7] Department of Life Sciences, Imperial College London, London, UK

Corresponding author
Daniel T.I. Bayley,
daniel.bayley.14@ucl.ac.uk

## ABSTRACT

Reef structural complexity provides important refuge habitat for a range of marine organisms, and is a useful indicator of the health and resilience of reefs as a whole. Marine scientists have recently begun to use 'Structure from Motion' (SfM) photogrammetry in order to accurately and repeatably capture the 3D structure of physical objects underwater, including reefs. There has however been limited research on the comparability of this new method with existing analogue methods already used widely for measuring and monitoring 3D structure, such as 'tape and chain rugosity index (RI)' and graded visual assessments. Our findings show that analogue and SfM RI can be reliably converted over a standard 10-m reef section (SfM RI = $1.348 \times$ chain RI—$0.359$, $r^2 = 0.82$; and Chain RI = $0.606 \times$ SfM RI + $0.465$) for RI values up to 2.0; however, SfM RI values above this number become increasingly divergent from traditional tape and chain measurements. Additionally, we found SfM RI correlates well with visual assessment grades of coral reefs over a 10 × 10 m area (SfM RI = $0.1461 \times$ visual grade + 1.117; $r^2 = 0.83$). The SfM method is shown to be affordable and non-destructive whilst also allowing the data collected to be archival, less biased by the observer, and broader in its scope of applications than standard methods. This work allows researchers to easily transition from analogue to digital structural assessment techniques, facilitating continued long-term monitoring, whilst also improving the quality and additional research value of the data collected.

## INTRODUCTION

The physical structure of coral reef habitats is a strong determinant of the abundance and diversity of many reef-associated organisms (*Graham & Nash, 2013*; *Darling et al., 2017*). Morphologically complex coral structures also indicate a reef's current health and its likelihood of rebounding from disturbance events such as heat-induced bleaching (*Alvarez-Filip et al., 2009*; *Graham et al., 2015*). Changes to reef structure can be ecologically relevant at a range of scales according to the reef's associated organism's body size; therefore, even centimetre-level changes in habitat can be important to reef community structure on a local scale (*Nash et al., 2013*).

Despite the importance of reef structural complexity and its incorporation into many standard reef-monitoring protocols (*Bayley & Mogg, 2019*), quantification and monitoring of structural changes through time has remained relatively simplistic. Typical monitoring budgets tend to be restrictive, so coarse visual or analogue methods that combine practicability with low cost—such as 'tape-and-chain rugosity' (*Risk, 1972*), broad qualitative visual estimation (*Wilson, Graham & Polunin, 2007*) or depth measures (*Dustan, Doherty & Pardede, 2013*)—are most commonly used. Structural assessments conducted using such methods tend, however, to be limited in scale due to SCUBA time restraints as they rely on researchers being in the water and are time-consuming to complete (*Knudby & LeDrew, 2007*; *Harborne, Mumby & Ferrari, 2012*).

Standard analogue techniques have proven useful for broadly describing reef structure for ecological analysis (*Alvarez-Filip et al., 2009*; *Graham & Nash, 2013*; *Graham et al., 2015*), but are criticised for being highly variable in their results due to recorder bias (*Wilson, Graham & Polunin, 2007*), and for often giving high variability from even small changes in measure placement. Furthermore, standard topographic measures such as 'rugosity' usually produce only one unbounded linear metric at a single coarse (centimetre) resolution, limiting the usefulness of such measures for describing complex differences in the physical form of individual underwater structures or reefscapes. The use of just one simplistic and poorly-repeatable metric is likely therefore to also limit our ability to relate reef structure to reef fish population sizes and community structure (*Knudby & LeDrew, 2007*; *Nash et al., 2013*; *Young et al., 2017*).

Recent advances in technology and computing power are providing new, data-dense and quantitative virtual techniques to measure the 3D structure of objects underwater. The use of LiDAR, Sonar, and satellite-based technologies to assess benthic structure and bathymetry is now commonplace (*Brown et al., 2011*), and has revolutionised the measurement of benthic topography, revealing new patterns and interactions in spatial ecology (*Brock & Purkis, 2009*; *Costa, Battista & Pittman, 2009*; *Purkis, 2018*). However, such methods are extremely costly to deploy, require specialist training to operate and are restricted by water depth. Furthermore they can only detect features greater than roughly one metre in size (*Kenny et al., 2003*; *Costa, Battista & Pittman, 2009*).

Land-based methodologies for accurate, cost-effective 3D measurement have been advancing rapidly and are now being adapted for use in aquatic environments. One such approach that has been gaining popularity is 'Structure From Motion' photogrammetry

(SfM) (*Westoby et al., 2012*), which creates scaled 3D digitally-derived virtual surface model renderings of objects in fine detail from multiple overlapping photographic or video images and reference markers. The imagery needed for such models can be collected using a single standard underwater camera, with no need for an expensive rig. Once the virtual reef surface has been created and calibrated using open-source or specialist software, detailed morphometric surface analyses can be undertaken on the object of interest.

Several studies have recently detailed the application of the SfM technique to underwater marine environments, showing the technique to be useful for quantifying structure across scales from colony to reef-scape (*Leon et al., 2015*; *Burns et al., 2015*; *Ferrari et al., 2016b*; *Teague & Scott, 2017*; *Young et al., 2017*). SfM has also been shown to be fast, accurate and repeatable (*Lavy et al., 2015*; *Burns et al., 2015*; *Figueira et al., 2015*; *Storlazzi et al., 2016*; *Bryson et al., 2017*). However, there has only been limited research into whether this new method of assessment can provide measurements that are directly transferable from current standard monitoring protocols (*Ferrari et al., 2016b*), meaning it is not yet clear whether ongoing surveys can transition to this new technique without risking the loss of comparability with older survey data.

Here, we empirically compare standard structural survey methods for coral reefs with SfM and test whether the resulting data can be inter-calibrated. We compare the SfM technique to the two most widely used standard methods of structure assessment: 'tape and chain rugosity' and graded visual assessment. We use a much larger set of validation transects than previous comparisons (*Ferrari et al., 2016b*; *Young et al., 2017*), and apply them over a range of coral reef habitats, at the most widely applied scale of 10 m. We also discuss SFM's additional possible outputs and its costs and benefits. We argue that SfM increases versatility, repeatability, and archival value of reef surveys.

## MATERIALS AND METHODS

### Data collection

Surveys were carried out at six locations in the same reef complex within the Danajon Bank double barrier reef, north of Bohol, Philippines. Surveys were conducted at depths of five to eight m in the daytime during November 2016, with horizontal visibility ranging from >10 to 5 m (Fig. 1).

Reef structure was recorded within varying visual grades of habitat complexity over a square 10 × 10 m (100 m$^2$) reef area following the method described by *Polunin & Roberts (1993)*, with grades ranging from zero to four, corresponding to: 0 = no vertical relief; 1 = low and sparse relief; 2 = low but widespread relief; 3 = moderately complex; 4 = very complex with numerous caves and fissures. For each survey area the dominant broad substrate type (i.e. >40% cover of sand/rubble/hard coral growth form/soft coral/algae) was also recorded visually (*English et al., 1997*).

Inside each 100 m$^2$ area, the rugosity index (RI) was calculated for 8–10 evenly-spaced parallel transects using the widely-used 'tape and chain' method (*English et al., 1997*). A 10-m brass chain with one-cm links was moulded to the hard substratum in order to measure the topographic (surface) distance. The direct horizontal distance between

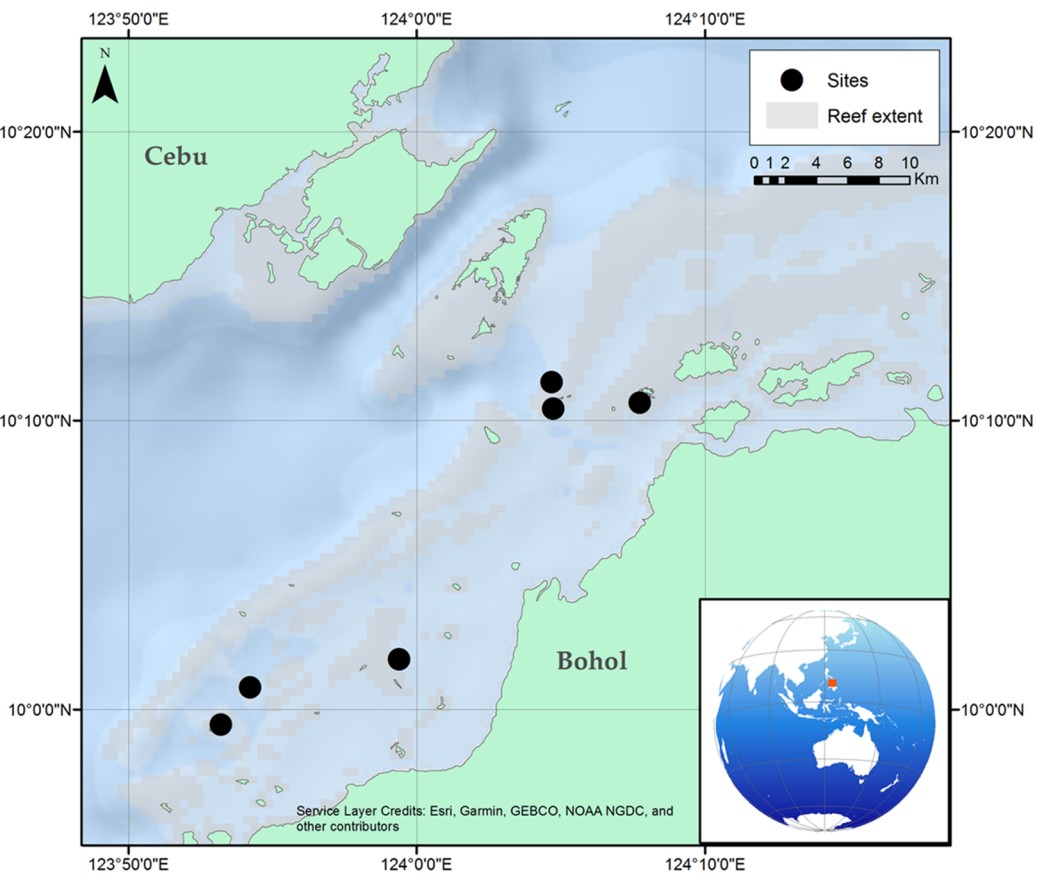

**Figure 1 Location of the six reef sites surveyed along the Danajon Bank reef complex, north of Bohol, Philippines.**

the start and end of the chain was then recorded, giving linear distance (Fig. 2). RI is then calculated as *Surface distance (SD)/linear distance (LD)*.

After each chain was laid, a weighted reflective marker of known dimensions (147 × 50 mm) was placed at the start and end of the chain and left in place while the chains were removed. This process resulted in a set of 20 fixed start and end markers in each quadrat, indicating LD for each transect (Fig. 2).

Following the in situ RI measurement, each whole quadrat was imaged across its full area in a lawnmower pattern from a distance of ~2 m above the substrate (dependent on visibility), following *Burns et al. (2015)*, to collect multiple overlapping images of the reef, using a Nikon D750 DSLR camera with a wide-angle fixed 20 mm Nikor lens and dome port. The same camera was used for all imaging, to capture high detail (6,016 × 4,016 px) images, and to prevent any potential variation between resulting models from differing camera or lens types (*Lavy et al., 2015*). The footprint of a typical photo in this study was approximately 2.5 × 1.75 m, (though this area can increase and decrease as the camera orientation shifts). Linear overlap of each photograph's footprint varied in line with substrate complexity, but ranged from 75% to 95% across this study. Sidelap (lateral overlap) ranged from 75%, up to 90%. Overlaps were calculated post hoc by placing markers in the middle of images and measuring the linear and lateral distances between image sets.

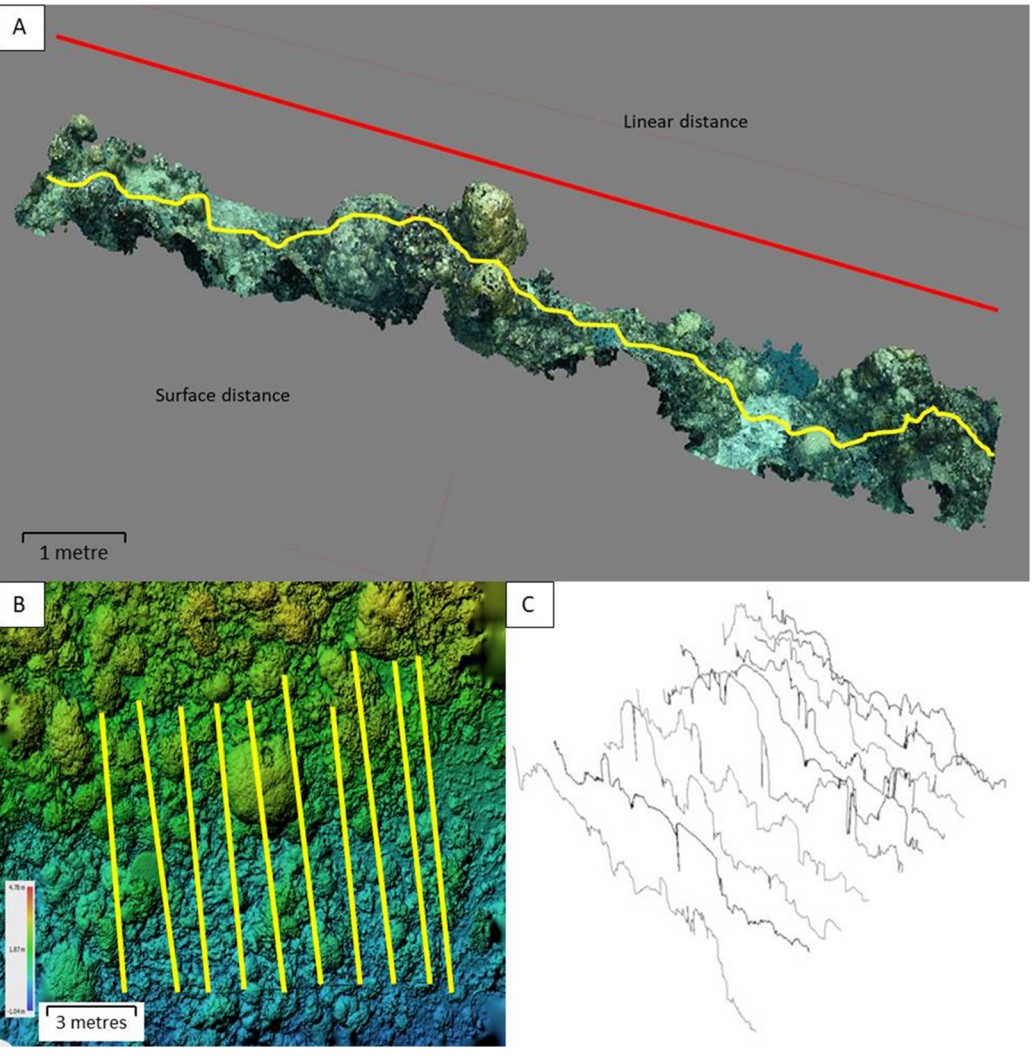

**Figure 2 A virtual reef with the rugosity index transects.** (A) Visualisation of linear and surface distance measurements across a typical 10 × 1 m cross-section of reef. (B) A DEM of a medium rugosity mixed growth form Philippine reef, illustrating the layout of a ~100 m² benthic quadrat with transects (scale and depth shown). (C) The 10 digital rugosity transects shown in section B, each with a 10 m surface distance.

Once images were collected, a digital surface model of the area photographed was created using Agisoft 'Photoscan' Professional (now Agisoft 'Metashape') software (Version 1.3.4; Agisoft, St. Petersburg, Russia; http://www.agisoft.com/downloads/installer/). Each ~100 m² model was based on the alignment of approximately 600–1,000 overlapping digital photographs, dependent upon light levels and structure of the benthic topography (alignment settings = high accuracy, generic pre-selection of images, 40,000 key-point limit, 10,000 tie-point limit). This process typically gave a dense XYZ point-cloud of around 4,000 matched points per m². The dense point-cloud was then converted into a Delaunay Triangulated Irregular Network wireframe mesh, with medium quality reconstruction, aggressive depth filtering and standard interpolation settings applied. Finally, using tiled imagery overlaid onto the filled reef mesh, 10 of the in-situ reflective
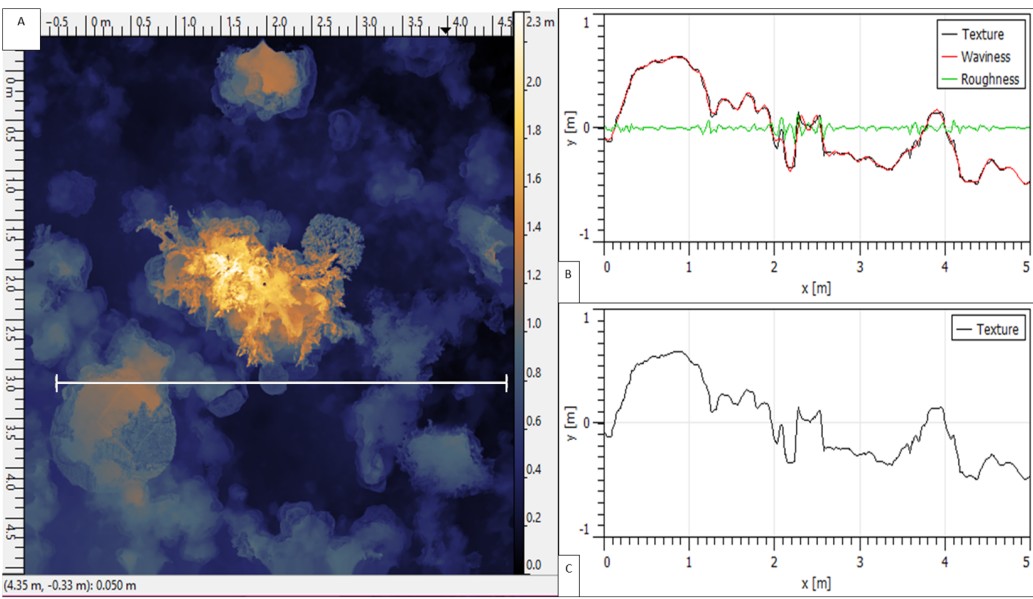

**Figure 3  A 25 m² reef area displayed within 'Gwyddion' software.** (A) XYZ pointcloud shown in false colour, with a white 5 m 'virtual transect' length selected. (B) 'Roughness', 'waviness' (at 0.2 m sampling frequency cut-off) and 'texture' of the virtual transect surface. (C) Overall texture (rugosity) of the transect—equivalent to 'Surface Distance' at 1 cm resolution.

markers' lengths were used to calibrate each model to these known distances (147 mm) in XYZ space, giving overall averaged scaling accuracies of <5 mm. Approximate depth (Z dimension) of the first marker in each quadrat was measured using a Suunto Gekko dive computer, and the models were orientated to a level XY plane using an in-situ spirit-level.

The XYZ pointclouds of each rendered reefscape were exported from Photoscan and analysed for RI metrics within Gwyddion software (*Nečas & Klapetek, 2012*), using the surface roughness analysis tool, following rasterization (at one cm pixel resolution, with averaged linear point interpolation, plus mirrored exterior), and SD measurement with zero cut-off (Fig. 3). Surface point values were averaged across 10 pixels to account for any fine-scale deviation of the in situ chain from the perfectly straight virtual transect line used in analysis.

## Data analysis

Linear regression was used to relate the matched LDs and RI values ($n = 58$) from in situ and virtual transects. If both methods give very similar RI estimates, the regression between them is expected to have an intercept of 0 and a slope of 1; Student's $t$-tests were used to assess whether the data fitted this expectation. We also used linear regression to relate the in situ and virtual measures of RI to visual reef estimation techniques (*Polunin & Roberts, 1993*) on a five point scale (0–4). Analysis of covariance was used to test whether the relationships between virtual and in situ RI varied among dominant substrate types. We explored log-transformation of RI estimates because, by inspection, variances were higher at higher values of RI; but transformation did not fully remedy this and made very little difference to
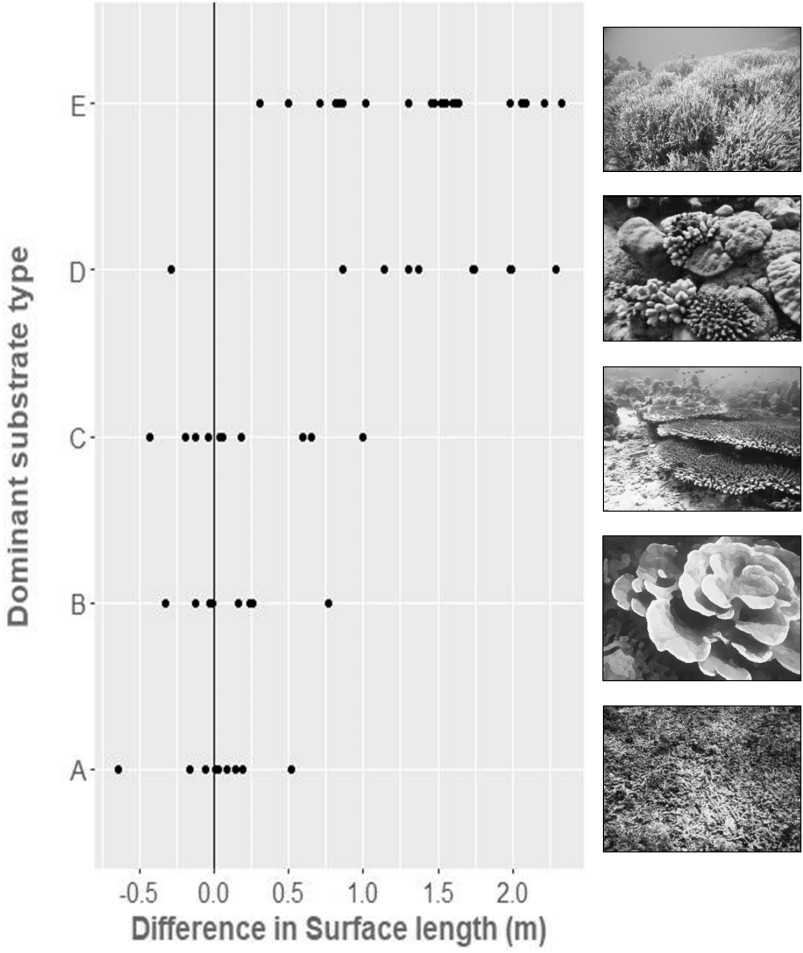

**Figure 4 The difference between in-situ chain measured surface distance (10 m) and virtually measured surface distance, within five broad substrate types.** (A) Flattened rubble, (B) foliose growth forms, (C) table and massive growth forms on sand, (D) sparse corymbose and massive growth forms, (E) dense branching coral growth forms.

goodness-of-fit, so we report the results from the untransformed data. All analyses were carried out using the lm() function in R, version 3.4.1 (*R Core Team, 2016*).

## RESULTS

### Tape and chain method

In situ (chain-measured) and virtual (SfM-derived) values of LD were practically identical ($R^2$ = 0.9994, mean difference = 0.002, Std. Dev = 0.02 m; all mean differences are reported as virtual—in situ measurement), as expected. Comparison of SD between the same points for the in situ chain-measured distance of 10 m and the virtually measured distance gave good agreement within simple substrates, that is, rubble fields (Mean difference = 0.01, Std. Dev = 0.29 m), foliose dominated reefs (Mean difference = 0.17, Std. Dev = 0.44 m), and mixed table/massive corals on sand (Mean difference = 0.12, Std. Dev = 0.33 m). There was, however, less agreement between the methods within more complex habitats, with higher variance and with virtual distances tending to be larger than in situ measurements

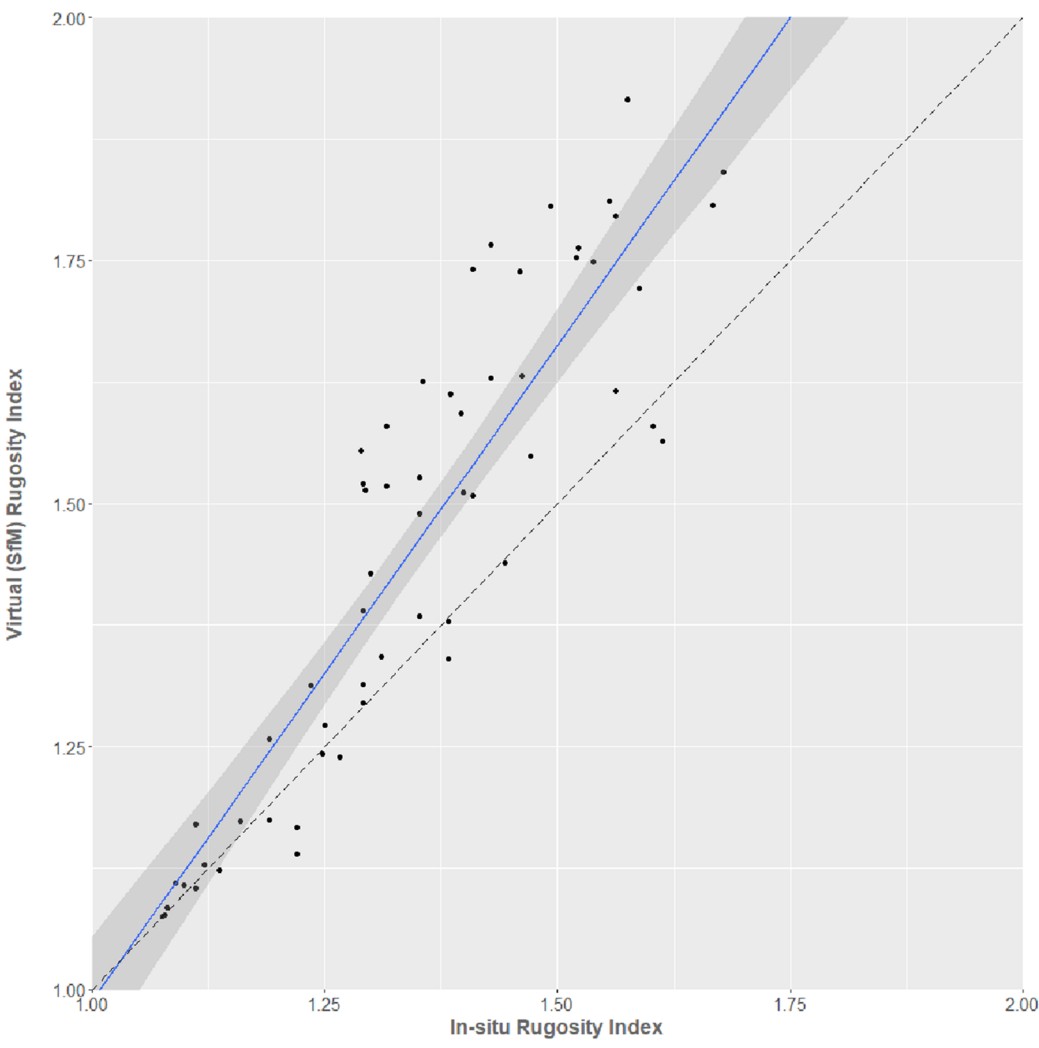

**Figure 5** **In situ chain-measured rugosity index against virtual SfM-derived rugosity index.** The blue line shows the linear relationship between the measures (95% Confidence Intervals shown). A dashed reference line (slope = 1, intercept = 0) is also shown for reference.

(Fig. 4), that is, dense branching *Acropora* thickets (Mean difference = 1.41, Std. Dev = 0.74 m), and mixed corymbose and massive growth form communities (Mean difference = 1.39, Std. Dev = 0.59 m).

Virtual RI was well-predicted by in situ SfM RI in the regression ($R^2$ = 0.82; Fig. 5). However, the intercept differed significantly from zero (estimate = −0.359, $t$ = −3.095, 56 df, $p$ = 0.003) and the slope differed significantly from 1 (estimate = 1.348, $t$ = 15.78, 56 df, $p$ < 0.001), because values for virtual RI gradually become greater for every corresponding value of in-situ RI as surfaces become more complex. The regression equations for converting between in situ and virtual estimates of RI are:

Virtual RI = 1.348 × (in situ chain RI) − 0.359

In situ chain RI = 0.606 × (virtual RI) + 0.465

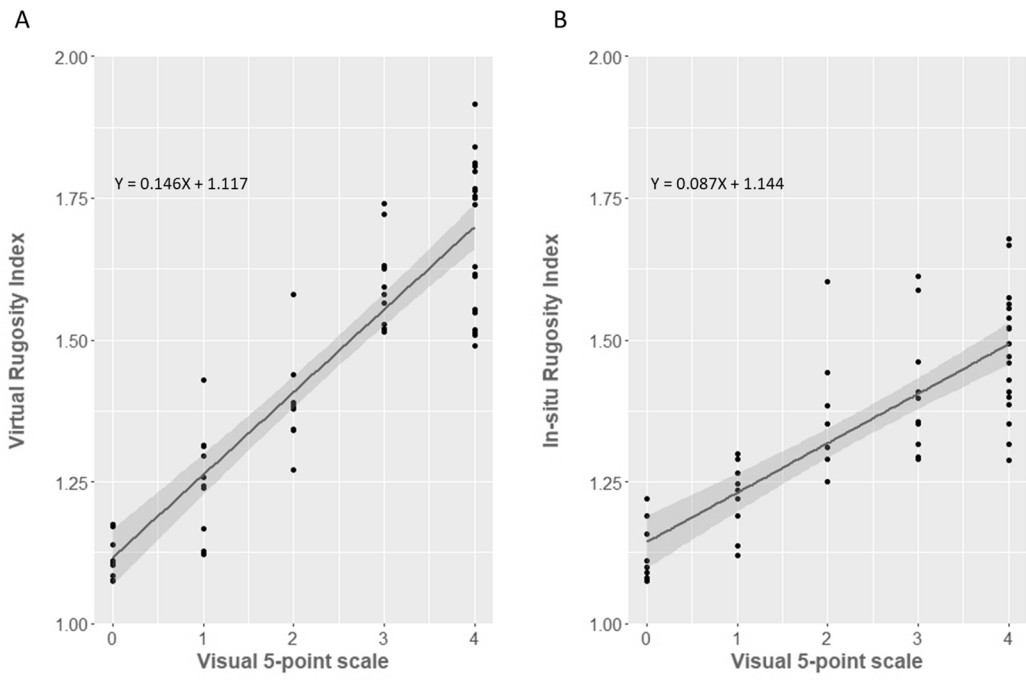

**Figure 6 Comparison between (*Polunin & Roberts, 1993*) visual method of structural assessment and the 'tape and chain'/SfM rugosity index methods.** (A) Virtual SfM rugosity method against the visual assessment method. (B) In situ tape and chain rugosity method against the visual assessment method. Individual data points overlaid and 95% Confidence Intervals shown in grey.

Inclusion of dominant substrate type as a factor increased the predictive ability of the regression (ANCOVA; $F_{6,51} = 104.6$, $p < 0.001$, $R^2 = 0.92$), demonstrating that the level of substrate complexity affects the accuracy of the RI comparisons.

### Visual estimation method

In situ and SfM-based methods both fit broadly to the visual assessment method proposed by *Polunin & Roberts (1993)* (Fig. 6). However, the fit was markedly better for the SfM virtual method ($R^2 = 83.1\%$, $F_{1,56} = 274.7$, $p < 0.0001$) than for the in-situ tape and chain method ($R^2 = 66.3\%$, $F_{1,56} = 110.4$, $p < 0.0001$). The regression equations for converting between virtual or in-situ estimates of RI and visual estimates are:

Virtual Rugosity Index $= 0.1461 \times$ visual grade $+ 1.117$

In-situ Rugosity Index $= 0.087 \times$ visual grade $+ 1.144$

## DISCUSSION

### Rugosity index assessment

Surface distances, and hence estimates of RI, produced by SfM photogrammetry over coral reefs correlated strongly with those from in situ tape and chain measurement, but SfM values were larger and more variable than tape and chain estimates in more structurally complex reefscapes. There are three likely reasons for the disparity in estimates

of SD (and therefore RI). Firstly, the models created through SfM become increasingly likely to produce areas of misaligned points as the complexity of the surface of interest increases, even with aggressive filtering of anomalous points from the dense point-cloud. These misaligned points lead to the need for greater post-processing cleaning for highly complex objects, or an expectation of some limited complexity overestimation.

The second reason for increased complexity values from the SfM derived method is because soft organisms, such as soft corals, extended polyp tentacles of hard coral, algae, worm feeding appendages and crinoids, all have structure which would not be recorded by the tape and chain method (*Ferrari, 2017*), but they will be recorded by SfM as long as these organisms are stationary during imaging. Relatively still water conditions are therefore needed for accurate assessment of soft structures such as these.

Finally, it is likely that the virtual surface created through SfM more accurately reflects the real structural complexity than in situ methods because an in-situ chain will typically fall off of finely branching structures and tend towards the lowest stable points of the reef through gravity. This makes measurement of slender, overhanging or highly complex objects difficult and likely to be under-estimated in terms of their rugosity, even if additional time is spent moulding the chain to complex/overhanging surfaces. The SfM technique shares this limitation of typically not being able to fully capture overhanging surfaces, however incorporating an increased number of oblique angle or upwards facing photos where possible (with appropriate lighting) will minimize this limitation and can allow vertical structures to fully overhanging cave systems to be imaged using SfM (*Hernández et al., 2016*; *Robert et al., 2017*; *De Waele et al., 2018*).

One shared limitation between the tape and chain and SfM methods for measurement of RI is an inability to fully assess structures such as densely branching growth forms and tightly packed overhanging structures such as terraced table corals. While SfM can capture more of this structure through oblique angled photography and good lighting, it is still limited by what can be seen, so obscured areas within a coral matrix for instance will still not be quantified (*Lavy et al., 2015*). Likewise, good underwater visual conditions are essential for adequate model creation as turbid, low light environments will produce lower quality outputs (*Bryson et al., 2017*).

## Visual complexity assessment

In a similar fashion to the comparison with tape and chain rugosity, SfM RI correlated very well to visual estimates of structural complexity over coral reefs, explaining 83.1% of the total regression variance. The SfM method furthermore improves our ability to compare visual complexity to RI values, with the visual estimates of structural complexity explaining 23% more of the variance from correlation to virtual (SfM) RI than from correlation to the in situ (tape and chain) RI. This indicates that SfM better matches how our eye naturally assesses complexity of structures. The reduced coefficient of determination using the chain method was primarily driven by the lack of differentiation in rugosity indices within higher complexity sites (two to four on the visual scale) and the wider variance of RI values at each grade. While no other studies that we know of directly compare SfM RI to visual assessments of structure, our results do match well with

**Table 1 Cost-benefits of four rugosity assessment techniques for analysing an area of 100 m$^2$ or for 10 (10 m) transects.**

| Technique | Field time | Processing time | Repeatable | Other analyses possible? | Remote assessment possible? |
|---|---|---|---|---|---|
| Tape and chain (× 10) | <1 h | None | No | No | No |
| Visual | <5 min | None | No | No | Yes |
| SfM | 30 min–1 h | ~1 day | Yes | Yes | Yes |
| SfM—cluster computing | 30 min–1 h | <1 h | Yes | Yes | Yes |

**Note:**
Cost-benefits of four coral reef structure assessment techniques for analysing an area of ~100 m$^2$ (using Structure from Motion (SfM) or visual techniques), or for 10 (10 m) in situ transects using the 'tape and chain' rugosity index technique. SfM processing time based on ~400 images within Agisoft Photoscan run with a 32GB RAM, Intel Core i7 processor, and NVIDIA GeForce GTX 960 graphical processor. The cluster used three nodes, each with the above stated specifications.

the study by *Wilson, Graham & Polunin (2007)*, which found visual estimates of reef topography were significantly correlated with tape and chain estimated rugosity.

Our analysis furthermore illustrates the pitfalls of taking single measures of RI, as relatively high variance of RI values are observed at each visual grade for both the in situ and virtual techniques. This shows the need for multiple replicates of linear RI to be taken at each site, whichever method is applied, in order to give a reliable approximation of overall structure.

## Costs, benefits, and limitations

The concept of 'photogrammetry' has been around as long as photography, but it is only within roughly the last 5 years that we have reached a point where advances in both digital camera technology and digital processing power have made the SfM technique economically viable and time-effective. The development of fast digital photogrammetry algorithms and the now widespread use of low-cost underwater action cameras such as GoPro$^{TM}$ opens up a plethora of opportunities in the marine world for recreating virtual benthic formations such as reefs, and enables us to analyse them in an efficient, objective and cost-effective way. These measures allow us to move from the simplistic chain or visual methods to objective and quantitative morphometric analyses of volume, surfaces, traits, and spatial distribution of individual colonies, which can be analysed at the scale of interest to the study. Work has indeed already begun to scale-up this method and integrate the use of this technology with drones to allow much greater area coverage with reduced risks and costs (*Chirayath & Earle, 2016*), which is likely to revolutionise shallow reef monitoring work.

Table 1 details the costs and benefits of each method discussed in this work, in terms of field and processing time, scope of analyses possible, and reproducibility of the analyses. The comparisons are not comprehensive but give a broad view of the most widely used methods contrasted against SfM using single or cluster-based processing. In this sense, a 'cluster' is a network of two or more computers acting as linked servers (nodes), essentially allowing greater processing power by combining the resources of each linked computer.

Whilst the time costs of initial data collection are comparable for each methodology, the initial processing time of SfM is the primary limiting factor currently, and is explicitly dependent upon the computer processing power available to the researcher. As processing time is reduced, through cluster-based processing, (or through likely future accelerated processing speeds as computing power advances), the range of added benefits from the SfM method become apparent.

Agisoft Photoscan is the software currently most commonly used for this type of 3D model generation of reefs, due to its easy interface, range of capabilities, relative low cost, and facility to integrate with workstreams such as Python. However, there are a range of other brands available such as Pix4D, Bundler and Autodesk, with differing specialities but broadly similar capabilities. Post-processing measurements can also be completed with a number of different software packages, dependent on the specified objective. Some frequently used packages are ArcGIS '3D analyst', Rhino, and the open-sourced Meshlab and R. We chose to use Gwyddion due to its versatility of functions, easy interface and because it is also open-source with good documentation. While Gwyddion was originally designed for electron microscope surface metrology, the techniques needed to analyse a reef surface are nevertheless the same.

There are of course still limitations to the SfM technique within a marine setting. For effective models to be created, the images collected need to be clear and sharp, and this can be a challenge in low-light/turbid/high energy environments. Steady camera work with adequate water clarity and good lighting is therefore essential. One of the benefits of SfM however is its ability to generate image mosaics far larger than those captured by an individual image. This means that, whilst it is possible to model objects of great size, it is also possible to model objects in very poor visibility, given sufficient image overlap. The relationship between the distance of the camera from the object and the resulting photographic footprint is a linear one, meaning that the footprint will be twice the size at twice the distance. In turn, this means that a survey conducted in poor conditions (e.g. 0.5 m effective visibility) will require four times the number of images to cover just the same photographic width as a single image from a survey conducted at two metres effective visibility. This resulting increase in photographic number has a knock-on effect on the computing power and resources need to model a single area, possibly rendering such an effort impractical in poor conditions with current general/mid-level computing systems. Conversely in good visibility conditions, the photographer must make a decision between coverage and detail, as image resolution will be lost the further the camera moves from the substrate (*Hitchin, Turner & Verling, 2015*).

Linked with the issue of adequate visibility is the need to bear in mind obscurement of objects, particularly in dense coral thickets, or from overlapping table growth forms (*Goatley & Bellwood, 2011*; *Figueira et al., 2015*). We were able to minimise the error associated with such habitats in this study by increasing the number of images taken in more complex habitats, by incorporating oblique angle photos (whilst ensuring to minimize large blue-water sections in the images), and by maintaining good lighting (*Hitchin, Turner & Verling, 2015*; *Burns et al., 2015*; *Pizarro et al., 2017*). Despite these considerations, some sections of reef, such as areas of dense coral matrix and highly

branching variable height sections where oblique images are restricted, will always have some level of obscurement that will introduce gaps to the model. While these gaps are interpolated in the model building process, they will be of course only be statistical estimations and this uncertainty must be considered and minimised whenever using this technique.

Following on from these stated limitations, this study investigates tropical coral reef habitats only, and not rocky or temperate biogenic reefs. While it is unlikely that any significant differences in the conversion between in situ and virtual measures of structure will occur in these differing habitats given the technique's accuracy (*Figueira et al., 2015*; *Bryson et al., 2017*; *Raoult et al., 2017*), this has not been explicitly tested here. We would therefore recommend future research across these different habitats and conditions, including further investigations into the effects of scale (i.e. colony to reefscape), and fractal dimension/resolution of measurement.

Despite these current limitations, our study demonstrates that the SfM technique can be used to easily transition from analogue to digital structural assessment, allowing continued long-term coral reef structure monitoring. Furthermore, the increased range of analyses available from the creation of virtual reefscapes is likely to lead to a clearer understanding of the ecological processes related to reef physical structure.

## CONCLUSIONS

The SfM technique shows great promise for future survey efforts due to its ease of use across multiple depths, scales, and reef types as well as its non-destructive nature. The outputs from a single survey can be used in a number of different analyses; for instance, virtual transects can easily be applied to currently laborious field methodologies such as estimation of carbonate budgets (*Perry et al., 2012*), increasing the speed, scale and accuracy of assessments. Additionally, the ability to measure not just hard corals but other often neglected aspects of the reef which provide important structure, such as soft corals or macro-algae, may help give insight into associated community dynamics.

Perhaps most importantly however, the SfM method is quantitative, is less biased by the recorder, and is replicable. The technique therefore allows detailed, spatially explicit observation of community change through time rather than typical purely qualitative descriptions (*Ferrari et al., 2016a*). Our ability to visualise and store the reef models will therefore allow verifiable and archival observations, an increasingly sought-after ability within the concept of reproducible science (*Munafò et al., 2017*).

### Funding

This work was supported by a Natural Environment Research Council (NERC) studentship and Zoological Society of London CASE award (NERC grant reference: NE/L002485/1). Additional funding for software and hardware used in analysis was provided through the NERC Services and Facilities capital equipment scheme. Equipment

support was provided by the Bertarelli Foundation. The funders had no role in study design, data collection and analysis, decision to publish, or preparation of the manuscript.

## Grant Disclosures

The following grant information was disclosed by the authors:

Natural Environment Research Council (NERC).

Zoological Society of London CASE award: NERC grant reference: NE/L002485/1.

NERC Services and Facilities capital equipment scheme.

The Bertarelli Foundation.

## Competing Interests

The authors declare that they have no competing interests. Andrew O.M. Mogg is employed by Tritonia Scientific.

## Author Contributions

- Daniel T.I. Bayley conceived and designed the experiments, performed the experiments, analysed the data, contributed reagents/materials/analysis tools, prepared figures and/or tables, authored or reviewed drafts of the paper, approved the final draft.
- Andrew O.M. Mogg authored or reviewed drafts of the paper, approved the final draft.
- Heather Koldewey contributed reagents/materials/analysis tools, authored or reviewed drafts of the paper, approved the final draft.
- Andy Purvis analysed the data, prepared figures and/or tables, authored or reviewed drafts of the paper, approved the final draft.

## Data Availability

The raw measurements are available in the Supplemental File, giving in situ and virtual linear length, surface length, visual grades and rugosity calculations for reefs across a range of complexity types.

## Supplemental Information

Supplemental information for this article can be found online at http://dx.doi.org/10.7717/peerj.6540#supplemental-information.

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
