# Peer review of "Capturing complexity: field-testing the use of ‘structure from motion’ derived virtual models to replicate standard measures of reef physical structure"

_PeerJ, doi:10.7717/peerj.6540_

## Round 0.1 · original submission · Minor Revisions

In addition to the reviewers comments below, I would like to read your rebuttal to the following comment I received from an expert that didnt have the time to do an in-depth review:

"Regarding rugosity, the failings of this purported metric have pointed out in the literature for more than 20 years. The problem in my opinion is that the rugosity concept is so deeply ingrained in reef ecology, that people use it without understanding that it is a biased, unbounded quantity. The following references are just a few that explain why rugosity is not a credible measure of topographic complexity.

McCormick, M. I., 1994. Comparison of field methods for measuring surface topography and their associations with a tropical reef fish assemblage. Marine Ecology-Progress Series, 112, 87–96

Commito, J. A., and Rusignuolo, B. R., 2000. Structural complexity in mussel beds: the fractal geometry of surface topography. Journal of Experimental Marine Biology and Ecology, 255(2), 133–152.

Frost, N. J., Burrows,M. T., Johnson,M. P., Hanley, M. E., and Hawkins, S. J., 2005.Measuring surface complexity in ecological studies.
Limnology and Oceanography-Methods, 3, 203–210."

So I would like you to consider inserting a comment in your manuscript of why despite this criticism, rugosity may have some usefulness for this field.

·

Basic reporting

Congratulations on a very clear, concise, well written and referenced paper.

It would be good to add a site overview figure showing the location of the reef complex and the six survey sites.

A few minor areas in need of re-wording/fixing are listed below:

Fix the following “Error! Reference source not found” messages: Line 110

Reword line 156 to remove slightly awkward phrasing “less good agreement”. Perhaps replace with: There was, however, less agreement between the methods within more complex habitats…

Reword lines 163-164. I think know what you mean by “virtual RI increases more quickly than in situ RI” but it is not technically correct as it implies some sort of speed component. I think it is a case of the virtual method measuring higher RIs than the in situ method particularly above a certain threshold.

Fix grammar in line 174.

Throughout the text alternate terminology is used for the same thing (e.g. virtual, digital, SfM etc… are used interchangeably; likewise, traditional, in situ, tape and chain are also used interchangeably). Be consistent as much as possible to make it easy for the reader.

Line 206 is a bit confusing. What is meant by “rugosity techniques if it is not SfM? Reword this sentence.

Line 209 change awkward wording “…in in situ…”

Line 219 change technological to technology.

Please explain cluster computer further and what a “node” (referred to in Table 1) is.

Experimental design

The methods are well-designed, clearly explained and replicable.

The following additions would be useful:
- In the data collection paragraph (lines 112-129) can you add an approximation of: how large the footprint of one photo was, how far apart the “lawnmower” tracks were, what the proportion of overlap (or overlap and sidelap) between adjacent images was?
- Please expand on lines 128-129. What exactly do you mean by “multiple known dimensions in xyz space”? Was z depth? Did you record depth at each marker? Did x and y dimensions come from the known size of the marker and/or the distance between markers? How was the model accuracy of <5 mm calculated?
- In lines 168-170 you say that dominant substrate type was included as a factor and increased the predictive ability of the regression. Please explain how dominant substrate type was included. Was each transect conducted over only one substrate type?

Validity of the findings

While the method presented is no longer novel the authors identify and fill an important niche and gap in our knowledge regarding how to articulate between traditional and SfM measures of rugosity to allow for comparability between surveys undertaken using the different methods. They also differentiate how their work differs from, and builds upon, previously published comparisons.

With respect to the discussion, has any other literature compared visual complexity assessment with SfM and tape and chain methods? If so, compare their findings in lines 205-216. If not, cool, make clear this is a novel aspect of your work.

Make sure you add to the conclusion your key finding which is that it is possible, using the regression relationships you found, to translate historical tape and chain RI measures into modern SfM measures, thus allowing long term monitoring studies to continue using the new method. It would also be good to add to the discussion whether your findings suggest that to do this a researcher should perform both methods at the same time (and if so how many replicates) to come up with a site-specific regression relationship or whether it may be possible for the same dominant substrate type to use a published regression relationship.

Additional comments

Congratulations on a nice paper, I look forward to reading the updated, published version!

·

Basic reporting

This paper is simple, but employs sound logic to address an important question: Are rugosity measures derived from photogrammetry compatible with those derived from chain-and-tape and visual estimate methods?

There is an opportunity to provide explicit and very useful management advice at the end of the discussion around how to validate and convert long-term monitoring data to photogrammetric records. An estimate of conversion factors to relate visual and chain-based complexity measures to future digital complexity measures (and variability around those factors) would add to the utility of this manuscript. Would reliably converting between digital and chain-based rugosity be possible? If not, which measures are responsible for the excess variability? This is not necessary but would add to the manuscript.

Inconsistency of terms (rugosity/RI/complexity and SfM/Photogrammetry/digital methods) creates confusion. This should be addressed throughout.

20. “Stereo-view” is a specific subset of photogrammetry (one that is not used in this paper). Probably omit. Also need to choose either SfM or photogrammetry as the term for the whole paper. Make clear that they are roughly equivalent, and then just use one term throughout.

28. This paper only deals with coral habitats. Not safe to imply that sfm = chain and tape in ALL situations. A brief discussion, backed up by literature, of how sfm is likely to function in rocky and other reef types would be useful here. This would allow broader inference to be drawn later.

61. LiDAR, sonar and satellite-based techniques are not more detailed (1 - 10m pixels). They ARE more data dense, archivable and cover greater extent.

62. LiDAR, not Lidar.

89. “However, there has been limited research into whether these new digital methods of assessment can provide measurements that are directly transferable from current standard monitoring protocols, meaning it is not yet known whether ongoing surveys can transition to this new technique without risking the loss of comparability with older survey data.” This is true, and this study does a good job of addressing this question (in coral habitats). But it feels deceptive to intentionally withhold the Ferrari 2016b reference (which does address this question) until after this statement of the core question.

110. Reference error.

119-121. Just specify resolution. By saying that settings and equipment was the same, you pre-empt the possibility of them effecting the resulting model.

134. “to account for any fine-scale movement of the in situ chain”. Unclear what this means. The chain is not modelled, correct? Why is fine-scale movement of chain be important?

188. Dense cloud, or sparse cloud? Which was filtered?

206. Unclear which rugosity. Digital or chain?

207-210. Confused sentence. Re-word to make clear. Start with what the heading says. “Visual estimates of complexity were more closely aligned with digital RI (have you introduced this term??) than with in-situ RI measurements. This indicates that digitally derived RI measurements better match how our eye naturally assesses complexity.”

Experimental design

The experimental design is simple, but dappropriate to the question. Replication is sufficient.

I feel that the novelty of the paper is perhaps over-stated as this question has been addressed previously (as acknowledged by the authors at the end of the introduction). Additionally, a key improvement of this paper over previous work on this topic is the increased replication, and repetition in “a range of reef habitats” (though this paper still only deals with coral-dominated substratum). This statement begs the question about rocky reefs, soft substratum, artificial substratum and the many other environments where rugosity is used. This should be reflected in the manuscript.

124-128. I recommend a table laying out the model building steps (align, dense cloud, mesh, etc), the settings chosen (high accuracy, medium density, aggressive filtering, etc), and the range of resulting features (1 million points, 790k faces, linear precision of <5 mm)

128. Although this study uses ratios (reducing the importance of scaling) scaling method should still be explained clearly. How were the markers used for scaling? I.e placed at known distances… And what were the scale features?

129. Many people will interpret this as model accuracy in 3D space, which it is not. Accuracy in XYZ implies that any point on the model is +/- # from other points in the model as related to the real world (see Figueira et al 2015 and Bryson et al 2017 for examples). Best to say scaling accuracy of <5mm. Also, how many used per model?

131. Make clear if each digital measurement was taken in the EXACT same line of reef as the physical chain, or if each is nearby and assumed to be representative. Either is fine, but unclear which was done.

Validity of the findings

This study meaningfully expands on some early exploration of digital rugosity's backward compatibility. This is an important question, which this work adds to. There is still much to be done in terms of different spatial extents and resolutions (of chain, and of digital reconstructions). Most importantly, it should be made clear that this work is on coral habitat only. Extrapolation to rocky and other reef types is probably valid, with some caveats. But it is not safe to imply that digital RI/chain RI relationships at all substrates are now known. The sample size, design and range of habitats covered here are not exhaustive.

193. This is true, but I wonder if sfm would capture these features either. Especially with limited resolution, any moving object is not likely to be reconstructed by sfm as it will not be in the same place in multiple images and will be stitched out by the Photoscan algorithms.

193-197. Confusing sentence. Good to speculate which method is closer to the real-world complexity. But the logic and sentence structure of reason #3 here needs improving. I also disagree with the logic. Sfm misses huge amounts of complexity in a coral reef as it does not model areas much beyond vertical. So all overhangs are lost or under-estimated. A well-deployed chain and tape method should capture these crevices as long as they are large enough to fit links of the chain into… irrespective of gravity.

204. Photogrammetry is a rapidly expanding tool in ecology and many people are interested in exploring it. If the authors have any guidance on specific limits (i.e. the authors found that having visibility at least 2x the distance from cameras to the substrate was sufficient to build accurate models) that would add to the usefulness of this manuscript. If not, it is not necessary.

Reviewer 3 ·

Basic reporting

Line 91 Previous studies that compared chain-and-tape rugosity to virtual rugosity include Ferrari 2016b and Young 2017.

Line 110 “Error! Reference Source not found.”

Line 156, improve writing by replacing “less good agreement” with “worse agreement”

Line 206 “SfM matches very well to visual estimates of structural complexity” Qualify this statement statistically.

Line 219 “very recently” is meaningless phrase here. Do you mean only in the past five years?

Sentence starting on line 259: strange use of semicolon. Use colons and parallel sentence structure. E.g., “… the SfM method is quantitative, less biased by the recorder, and replicable. It allows detailed spatially explicit…”

Line 263: Why is reproducible science in quotes?

Table 1: (A) Processing time is meaningless unless you know the computing power. These are approximate, so you could add in caption something like “Processing time is approximate for a standard laptop with 8Gigs Ram, etc.” (B) Consider removing the cost column. Many SfM software options are free. Visual surveys have the dive time and equipment as a cost, as do these other methods. Why’d you decide to remove that cost? How does a chain and tape cost £100? What areas are these covering? At the very least fix capitalization of Software/Kit Cost title, although kit doesn’t not include dive kit, so again I’d just delete this column to avoid confusion and because it doesn’t add much value to your main findings. (C) What does archival mean in this context? I think the word “Repeatable” covers what you mean here. You can archive the data recorded in Visual assessments by recording your measurements; so I don't think archive captures what you mean here.

Caption Figure 1: Capitalize sentence in part (A). Spell-out numbers that start a sentence (in part C).

Figure 1: Great figure, but please add scale. How many meters long is the linear distance? The scale in part (B) is too small to read. Do the colors mean anything in (C)? If not, I suggest removing colors.

Experimental design

Thanks for providing the raw data, but data labels are not clear. What is SR vs LR vs RR? How did you define the dominant substrate?

Methods: Please describe how you determined dominant substrate.

Validity of the findings

Nice work.

Additional comments

This is a nice paper that adds to a growing body of work. I recommend for publishing with minor comments detailed above.

---

## Round 0.2 · accepted · Accept

I have reviewed the revised manuscript and your rebuttal document and I believe you have satisfied both the reviewers and my concerns. Two minor comments:

1) There a few typos left (e.g., spaces missing between words), so make sure you do a final proofread of your manuscript.
2) While you mention the need for multiple replicates at each site, it would be interesting to point at knowledge gaps to assess s the stability of the SfM indices under different conditions (turbidity, illumination) acquisition parameters (camera specs, distance to target, overlap, etc), and SfM processing software and parameters.
Kudos again

#